# The Catalytic Subunit of *Schizosaccharomyces pombe* CK2 (Cka1) Negatively Regulates RNA Polymerase II Transcription through Phosphorylation of Positive Cofactor 4 (PC4)

**DOI:** 10.3390/ijms23169499

**Published:** 2022-08-22

**Authors:** Diego A. Rojas, Fabiola Urbina, Aldo Solari, Edio Maldonado

**Affiliations:** 1Instituto de Ciencias Biomédicas (ICB), Facultad de Ciencias de la Salud, Universidad Autónoma de Chile, Santiago 8910132, Chile; 2Programa de Biología Celular y Molecular, Instituto de Ciencias Biomédicas (ICBM), Facultad de Medicina, Universidad de Chile, Santiago 8380492, Chile

**Keywords:** phosphorylation, CK2, transcription, general transcription factors (GTFs), RNA polymerase II, *Schizosaccharomyces pombe*

## Abstract

Positive cofactor 4 (PC4) is a transcriptional coactivator that plays important roles in transcription and DNA replication. In mammals, PC4 is phosphorylated by CK2, and this event downregulates its RNA polymerase II (RNAPII) coactivator function. This work describes the effect of fission yeast PC4 phosphorylation on RNAPII transcription in a cell extract, which closely resembles the cellular context. We found that fission yeast PC4 is strongly phosphorylated by the catalytic subunit of CK2 (Cka1), while the regulatory subunit (Ckb1) downregulates the PC4 phosphorylation. The addition of Cka1 to an in vitro transcription assay can diminish the basal transcription from the Ad-MLP promoter; however, the addition of recombinant fission yeast PC4 or Ckb1 can stimulate the basal transcription in a cell extract. Fission yeast PC4 is phosphorylated in a domain which has consensus phosphorylation sites for CK2, and two serine residues were identified as critical for CK2 phosphorylation. Mutation of one of the serine residues in PC4 does not completely abolish the phosphorylation; however, when the two serine residues are mutated, CK2 is no longer able to phosphorylate PC4. The mutant which is not phosphorylated is able to stimulate transcription even though it is previously phosphorylated by Cka1, while the wild type and the point mutant are inactivated by Cka1 phosphorylation, and they cannot stimulate transcription by RNAPII in cell extracts. Those results demonstrate that CK2 can regulate the coactivator function of fission yeast PC4 and suggests that this event could be important in vivo as well.

## 1. Introduction

The coactivator PC4 (Sub1 in *S. cerevisiae*) was first identified from the upstream factor stimulatory activity (USA) fraction in HeLa nuclear extracts as a coactivator for RNAPII transcription in DNA templates without chromatin [1,2]. This 15 kDa protein possesses two short domains rich in serine and acidic residues which were named SEAC motifs and a DNA binding domain in the carboxy-terminal region [3]. This protein has multiple functions in transcription, such as a being a positive cofactor for transcriptional activation; facilitating promoter escape; interacting with the preinitiation complex in basal transcription; binding to dsDNA and participating in replication, DNA repair and cellular growth [4,5,6]. It has been demonstrated that PC4 is associated with chromatin and is important for cell cycle and chromatin condensation [7]. Phosphorylated PC4 negatively regulates the transcription in a VP16 activator-dependent system, and it has been described that phospho-PC4 loses its coactivator properties [8]. On the other hand, PC4 has been described as a new member of the transcription machinery of promoters containing an initiator (Inr) and a downstream promoter element (DPE) [9]. In the model of the Inr-DPE promoter, PC4 participates by positively regulating the transcription, but its phosphorylated form negatively regulates the transcription, since the phosphorylation event inhibits its coactivator function. One of the protein kinases described as responsible for PC4 phosphorylation is CK2, which negatively affects the function of this coactivator [5,8,10]. Fission yeast PC4 has also been described and it is encoded by the Sub1 gene, and the recombinant polypeptide can work together with TFIIA as cofactors for transcriptional activation [11].

Casein kinase 2 (CK2) is a Ser/Thr protein kinase that recognizes Ser or Thr in an acidic context, where the canonical motif is S/TXXD/E [12,13]. This protein kinase is present and expressed in all eukaryotic cells and it is located in the nucleus and cytoplasm. CK2 is an evolutive conserved protein kinase as judged by the high identity of the orthologues from yeast to mammalian cells, and mammal CK2 can be exchanged in yeast cells [14]. This enzyme is heterotetrameric, with two α and/or α’ subunits (CK2α or CK2α’) and two β subunits (CK2β), where the CK2α/α’ complex contains the catalytic activity and is active by itself, while CK2β is a regulatory subunit. This regulation may be positive or negative depending on the kinase substrate [15]. Both CK2α and CK2α’ are encoded by two different genes [16]. The CK2β subunit participates in the substrate specificity and the stabilization of the complex enzyme–substrate [17]. The three-dimensional structure of the CK2 holoenzyme has been described by site-directed mutagenesis and X-ray crystallography [18,19,20,21]. These data showed the importance of a zinc-finger motif for the dimerization of CK2β [18] and a preformed conformation of CK2β in the interaction with CK2α [22]. In addition, the structure of the holoenzyme shows a singular interaction between the regulatory subunit and the N-terminal domain of the catalytic subunit. Another characteristic of the catalytic subunit is the mimetic of GTP plus water by ATP in the active site of CK2α, indicating a “dual-co substrate specificity”, with the water molecule being the switch for the exchange of substrate in the active site of CK2α [21].

CK2 can regulate several cell functions such as signal transduction, cell cycle and gene expression, and also can phosphorylate and regulate the activity of substrates such as cytoskeletal and tumor suppressor proteins [23,24,25]. In the context of gene expression regulation, nuclear proteins have been described as CK2 substrates and the phosphorylation is positively regulated by CK2β, such as general transcription factors (GTFs) and specific transcription factors, suggesting that CK2 might play a role in gene expression regulation [26,27,28,29,30].

Despite PC4 functions being widely described in mammalian cells as described previously in the first paragraph, the yeast PC4 has not been well studied in vitro yet. A yeast orthologue of human PC4 was purified and named Tsp1 [31]. It is unknown whether Tsp1 is phosphorylated or not by CK2 in budding yeast and what the effects are of such modification, although it has been demonstrated that it can be phosphorylated and the phosphorylation can modulate the transcriptional activity of Tsp1 [31]. Dephosphorylated Tsp1 can interact with TFIIB, while the phosphorylated form cannot. Likewise, the dephosphorylated form can interact with the Gal4-VP16 activator, whereas the phosphorylated Tsp1 interacts less tightly [31]. On the other hand, dephosphorylated Tsp1 binds strongly to ssDNA, while the phosphorylated form binds weakly to ssDNA [31]. In budding yeast, an orthologue of PC4 was identified as a suppressor of the cold-sensitive phenotype caused by mutations in TFIIB (Sua7) and named Sub1, which is identical to Tsp1 [32]. It is known that yeast Tsp1/Sub1 can act as a coactivator for RNAPII in yeast transcription assays, as well as that fission yeast PC4 can stimulate basal and activated transcription in TATA-Inr and TATA-less promoters, a function which is dependent on TFIIA [11].

In this work, we describe the effect of fission yeast PC4 phosphorylation by CK2 over the basal transcription in an in vitro approach using fission yeast cell extracts, a system which closely resembles the cellular milieu, since it contains most of the factors present in the cell. To address this, we expressed both subunits of CK2α and β from *Schizosaccharomyces pombe*, Cka1 and Ckb1. We found that Cka1 strongly phosphorylates fission yeast PC4, which is negatively regulated by Ckb1. In in vitro transcription assays using PC4 and CK2, we found that Cka1 diminishes basal transcription in a TATA-Inr containing promoter as the template; however, Ckb1 and recombinant PC4 recovered and stimulated the transcription, indicating the role of CK2 in transcription via PC4 modification. In addition, the CK2 phosphorylation sites were identified in fission yeast PC4, and it was shown that a mutant in serine residues is not inactivated by phosphorylation by CK2 and it can activate basal transcription after in vitro phosphorylation by CK2.

## 2. Results

### 2.1. Protein Expression of CK2α and CK2β from S. pombe and Casein Phosphorylation

The nucleotide and amino acid sequences of CK2α (Cka1) and CK2β (Ckb1) from *Schizosaccharomyces pombe* are available in the Genbank database of the NCBI. Using that information, we cloned the genes for both proteins in an expression vector pET15b containing a histidine-tag (His-tag) at the N-terminus of each polypeptide. The proteins were expressed in bacteria and purified from the inclusion bodies. At this point, we used a protocol that includes a renaturation step in which the glycerol concentration in the dialysis buffer is critical (see Materials and Methods). In the case of Cka1 we could not renature the protein with 10% glycerol, and we used 30% glycerol in the dialysis buffer to obtain a suitable amount of renatured protein ready for Ni-NTA-Agarose chromatography. Recombinant proteins were analyzed in PAGE-SDS followed by silver staining (Figure 1A). The molecular weights determined from the gel were 49 kDa for Cka1 and 35 kDa for Ckb1, which were according to the expected size from the amino acid sequence. In the case of Cka1, we observed two bands in the silver-stained gels. Both polypeptides correspond to Ck1α, as they react with antibodies raised against *Xenopus* CK2α in a Western blot (Figure 1A). In the case of Ckb1, we detected the recombinant protein as a unique band in PAGE-SDS gels followed by silver staining (Figure 1A). The function of Cka1 was analyzed through phosphorylation assays using [γ^32^P] ATP and the classic CK2 substrate—casein. In order to determine the kinase activity, we performed Cka1 titration experiments. Phosphorylation of the substrate casein was determined in a dose-dependent manner (data not shown). The optimal enzyme amount for posterior assays was 2 pmol. To evaluate the role of Ckb1, increasing amounts of this protein were added to the assay containing Cka1 and casein (Figure 1B). When only Cka1 and casein were present in the assay, we assumed basal activity of the enzyme (Figure 1B, lane 2). Cka1 by itself can weakly phosphorylate casein (Figure 1B, lane 2); however, the phosphorylation of casein augments when increasing amounts of Ck1b are added to the assay containing a fixed amount of Cka1 (Figure 1B, lanes 3–5 and Figure 1C). Higher amounts of Ckb1 can inhibit the casein phosphorylation (Figure 1B, lane 6 and Figure 1C). All lanes in Figure 1B contained equal amounts of casein substrate, as it can be seen from a PAGE-SDS gel stained with Coomassie Blue R-250, which is shown in Figure 1D (lanes labeled casein). In summary, these results indicate that Cka1 possesses casein kinase activity, which is stimulated by the regulatory subunit Ckb1. In addition, the results found were comparable to those obtained in our laboratory with *Xenopus laevis* CK2 with 75 mM KCl as the optimal salt concentration (data not shown). The *X. laevis* CK2 has been characterized elsewhere [33].

### 2.2. Coactivator PC4 Is Phosphorylated by Cka1 and Negatively Modulated by Ckb1

One of the multiple functions of CK2 is the regulation of gene expression through the phosphorylation of different transcription factors. In mammals, one of these substrates is the transcriptional coactivator PC4, which has been described as undergoing CK2-dependent phosphorylation at the N-terminal region. In our laboratory, we have cloned *S. pombe* PC4 (fission yeast PC4) and we have described its transcriptional coactivator functions [11]. The sequence of this protein was analyzed using bioinformatic software (NetPhos3.1). Interestingly, we found two putative CK2 phosphorylation sites in the carboxy-terminal region. These residues are S98 and S100 (Figure 2A). In order to determine whether fission yeast CK2 can phosphorylate PC4, we performed in vitro kinase assays, first by adding only Cka1 and fission yeast PC4. We found that the level of fission yeast PC4 phosphorylation increased according to the amount of Ckb1 added (Figure 2B,C). We observed a peak of 60-fold increase using 4 pmol of Cka1 (Figure 2B,C, lane 7). This experiment was replicated using *Xenopus laevis* CK2α and the same results were obtained (not shown). Previously, heparin has been described as a competitive inhibitor of CK2 activity [34], and phosphorylation assays in the presence of heparin could indicate that the protein kinase present in the assay is a CK2 enzyme. Inhibition assays were performed adding heparin in increasing amounts to the in vitro kinase assay. We found inhibition of PC4 phosphorylation when 0.25 μg/mL of heparin was added to the assay. Indeed, the enzymatic activity decreased to 20% compared with an assay without heparin (Figure 2D, lane 3). PC4 phosphorylation was completely abolished when 4 μg/mL of heparin was added to the assay (Figure 2D,E, lane 7).

Next, we sought to study the role of Ckb1 in the catalytic modulation of Cka1; therefore, we added increasing amounts of Ckb1 to the phosphorylation assay containing PC4. Interestingly, we found that Ckb1 decreased the Cka1-dependent PC4 phosphorylation. The catalytic activity was decreased about 50% when 0.5 pmol of Ckb1 was added to the assay (Figure 3A,B, lane 4), compared with the assay without Ckb1. Increasing amounts of Ckb1 greatly diminish PC4 phosphorylation (Figure 3A,B, lanes 5–7). With 2 pmol of Ckb1, the activity was completely abolished (lane 7, Figure 3A,B). These results were compared with a control experiment using recombinant CK2 from *Xenopus laevis* and fission yeast PC4 as a substrate. Interestingly, PC4 is highly phosphorylated by *Xenopus* CK2α, and CK2β can negatively regulate the catalytic activity of *Xenopus* Ck2α on fission yeast PC4 (Figure 3C,D), although the levels of regulation were different compared with fission yeast Ck2a (compare Figure 3C with Figure 3D).

In summary, these results indicate that Cka1 strongly phosphorylates fission yeast PC4 and Ckb1 negatively modulates the catalytic activity of Cka1. In addition, the catalytic activity modulation of CK2α over PC4 phosphorylation by CK2β was also observed when recombinant *X. laevis* CK2 was used.

### 2.3. Ckb1 and PC4 Recover the Transcription Downregulation by Cka1

We studied the effect of CK2 in in vitro transcription on a promoter containing TATA and Inr called pG5-MLP, which contains the TATA box and Inr from the Ad-MLP promoter fused to five Gal4 binding sites [11,35]. When increasing amounts of Cka1 were added to the assay, we found inhibition of the basal transcription (Figure 4A,B, lanes 3 and 4 compared with lane 1) around 50–60%; however, Ckb1 alone had no effect on the basal transcriptional activity (Figure 4A,B, lane 5 and 6). Interestingly, when increased amounts of Ckb1 were added to the assay containing a constant inhibitory amount of Cka1, we observed the recovery of the basal transcriptional activity (Figure 4A,B, lanes 7–9). These results suggest that CK2 might have a role in transcription by RNA polymerase II in fission yeast.

The role of PC4 in in vitro transcription has been described in human cells and also in *S. pombe* [11]. In this study, in vitro transcription assays were used to study the effects of Cka1 and Ckb1 on fission yeast PC4. The basal level of transcription (Figure 4C,D, lane 1) did not change when TFIIEβ was added to the assay (Figure 4C,D, lane 3). However, the addition of PC4 to the assay showed a 2.5-fold increment (Figure 4C,D lane 4). Then, we evaluated the role of CK2 in the basal transcription stimulation of PC4; to this end, transcription assays were performed only with Cka1 or Cka1/TFIIEβ, showing transcription inhibition of 70% (Figure 4C,D, lanes 5 and 6). Interestingly, when transcription assays were performed with Cka1 and increasing amounts of PC4, stimulation of the basal transcription was augmented to a 4-fold increment (Figure 4C,D, lanes 7, 8 and 9). These results indicate that PC4, described as a coactivator, recovered the downregulation of the transcription when Cka1 was added to the assay. It is important to mention that PC4 at high levels is able to inhibit transcription stimulation (compare lanes 8 and 9, Figure 4C). This effect could be due to a squelching of transcription factors, since PC4 is able to interact with the general transcription factors. Alternatively, it might be due a repression in which high amounts of PC4 could result in non-specific binding to dsDNA, and this might compete with the binding of the general transcription factors necessary to form a preinitiation complex on the promoter [4].

### 2.4. Serine Residues S98 and S100 in PC4 Are Important for CK2 Phosphorylation

In an effort to determine the critical residues in fission yeast PC4 which are phosphorylated by CK2, residues S98 and S100 were changed to alanine. Point mutation of S98 reduces the PC4 phosphorylation by almost 30%; however, when both serine residues were simultaneously changed to alanine, the phosphorylation activity of PC4 by CK2 dropped to 0%, indicating that S98A and S100A are the in vitro phosphorylation sites for CK2 (lane 4, Figure 5A,B). The decrease in phosphorylation level observed in PC4 and PC4 mutants was not due to reduced levels of proteins in the assays, as is indicated in the Western blot analyses (Figure 5A, bottom panel).

Next, we addressed the functional consequences of the above mutations in the in vitro transcription assay using the Ad-MLD template. The wild type, S9A8 and S98A/S100A mutants were in vitro phosphorylated by fission yeast Cka1, purified and then added to the transcription assay. Unphosphorylated S98A and S98A/S100A mutants were active to stimulate transcription as well as the wild type PC4, indicating that the mutations per se do not inactivate its function (Figure 5C,D, lanes 1, 6 and 7). Inactivated wild type PC4 at 100 C, or addition of a heterologous protein such as bovine serum albumin (BSA), does not stimulate transcription further than PC4 itself (Figure 5C, compare lanes 1, 3 and 4). BSA by itself does not stimulate transcription (Figure 5C, lane 10), indicating that transcription stimulation is due to an active PC4 protein. As expected, the wild type enzyme phosphorylated PC4 and the S98A mutant were not able to stimulate transcription (Figure 5C,D, lanes 8 and 9), while the double mutant was active to stimulate transcription after incubation with CK2 and ATP, indicating that it is no longer regulated by CK2 phosphorylation (Figure 5C,D, lane 11). The effect of Ckb1 on the negative regulation of Ck1a on fission yeast PC4 was also investigated. Wild type PC4 and mutant S98A were phosphorylated in vitro in the presence of the holoenzyme (Cka1/Ckb1) plus ATP, and afterwards PC4 proteins were purified to separate the CK2 holoenzyme and added to the transcription assay. Wild type PC4 and mutant S98A were active after the treatment with the holoenzyme (Figure 5E,F, lanes 2 and 7), indicating that in the presence of Ckb1 both wild type PC4 and the S98A mutant were not phosphorylated to an extent to inactivate the transcriptional stimulation function. On the contrary, the treatment of PC4 with Cka1 alone is able to abolish the transcriptional stimulatory activity of both wild type PC4 and mutant S98A (Figure 5F, lanes 3 and 4). The double mutant S98/S100 was not affected by the treatment with the CK2 holoenzyme plus ATP (Figure 5F,G, lane 8). It has been reported that phosphorylation of human PC4 by CK2 inhibits its dsDNA binding activity and therefore inhibits the transcriptional stimulation activity of PC4 [4,10]. We investigated the effect of phosphorylation of PC4 by Ck1a on the dsDNA binding activity of this coactivator. Recombinant PC4 binds to the dsDNA labeled probe; however, the increasing of phosphorylation by increasing the amount of Cka1 inhibits its binding activity (Figure 5G, lanes 2–5), but similar treatment on the mutant S98A/S100A does not have such inhibitory effect (Figure 5G, lanes 6–8).

## 3. Discussion

Mammalian PC4 is a small multitalented RNA polymerase II coactivator, which was identified by its ability to stimulate reconstituted activated transcription in vitro [8]. Later, it was identified in yeast as the Sub1 gene by its ability to suppress the cold-sensitive phenotype of TFIIB mutations. It was also identified as Tsp1 by its ability to stimulate basal transcription by RNA polymerase II [31]. The Sub1/Tsp1 gene product is larger than PC4, since it possesses an extra carboxy-terminal region of 190 amino acids, with no functions assigned yet; however, it could be an additional regulatory region and Sub1 might have some functional differences with PC4 as a consequence of this additional region. Calvo and colleagues have proposed that the extra carboxy-terminal region of Sub1 is necessary for protein stability, and it might regulate its DNA binding activity and also could interact with the Rpb 4/7 heterodimer of RNA polymerase II [6,36]. They have proposed that the extra carboxy-terminal region could modulate the DNA binding activity by in vivo phosphorylation, since that region contains phosphorylated residues [6,37]. Sub1 is not essential for yeast cell viability, since deletion of the Sub1 gene does not affect viability; however, it produces inositol auxotrophy, a phenotype associated with mutations in other transcription factors [32]. By using the model of PC4 knockout in mice, it has been found that loss of PC4 results in early embryo lethality, indicating an essential role of PC4 in embryonic development [38]. In addition, conditional knockout mice models showed defects in the development of the nervous system [38] or impairment of inflammatory response [39]. In addition, an overexpression of PC4 results in the malignant transformation of normal cells in several cancer models, indicating that PC4 has a role in tumorigenesis [40,41,42].

Sub1 has been described as a component of the preinitiation complex (PIC) [43]. Moreover, Sub1 can influence the transcription elongation rate and thus splicing [36]. In addition, Sub1 modulates transcription termination by interaction with the 3′ end processing factor Rna15 [6]. Additionally, Sub1 influences CTD phosphorylation through the transcription process [44]. In addition, Sub1, similar to mammalian PC4, has a role in transcription by RNA polymerase III [45,46], and as a DNA binding protein has also been implicated in DNA repair and the maintenance of genome stability [47,48]. Mammalian PC4 has been demonstrated to have a role in replication, chromatin condensation and cell cycle progression through Aurora Kinases [49].

Fission yeast also possesses a PC4 orthologue which is similar in size and function to mammalian PC4. Fission yeast PC4 is 136 amino acids long, compared to mammalian PC4, which is 127 amino acids long, and they share high identity along the entire sequence of the polypeptide. Similar to budding yeast Sub1, fission yeast PC4 can bind single- and double-stranded DNA and it is able to mediate transcriptional activation together with TFIIA and Mediator [11]. Fission yeast PC4 is also required for in vitro transcription of ribosomal protein genes [50] and for transcription of promoters containing an Inr as the unique core promoter element [51]. However, fission yeast PC4 is not essential for cell viability (deletion mutant) and its absence does not cause any visible phenotype [52]. This indicates that PC4 is required for transcription of a small set of non-essential genes in fission yeast or alternatively has a redundant function with other transcriptional coactivators.

Mammalian PC4 has been shown to be negatively regulated by phosphorylation [8]. It has been demonstrated that human PC4 is phosphorylated in vitro as well as in vivo by CK2, and the phosphorylation is restricted to seven serine residues located at the N-terminal region of PC4 in two serine and acidic residue-rich (SEAC) domains [8]. The phosphorylation of those SEAC domains inhibits the binding to the VP16 activation domain and also inhibits the ability to mediate transcriptional activation, since phosphorylated PC4 does not bind dsDNA [8,10]. Using the NetPhos3.0 program, we analyzed the potential phosphorylation sites on fission yeast PC4 and we found several potential phosphorylation sites. Phosphor-sites were found at the N-terminus of the molecule, and they are predicted to be phosphorylated by protein kinase C (PKC). Whether those sites can be phosphorylated by PKC is still unknown. We focused our analysis on CK2 phosphorylation sites, and it was found that Ser96, Ser98 and Ser100 are predicted to be phosphorylated by CK2. Indeed, those sites are phosphorylated by CK2 in vitro, since mutations of both S98 and S100 residues completely abolish the in vitro phosphorylation by CK2. Moreover, those sites seem to be important for CK2 regulation of PC4 function because phosphorylation of those sites by fission yeast Cka1 negatively regulates the transcription stimulatory activity of PC4. Additionally, PC4 phosphorylation inhibits its dsDNA binding activity, which is important for the coactivator function. In this regard, we suggest that phosphorylation might be able to affect both constitutive and induced transcription in vivo. Location of the phosphorylated residues by CK2 is different in human PC4 compared to fission yeast PC4, since in human PC4 they are located at the N-terminus, whereas in fission yeast PC4 they locate at the C-terminus (see Figure 2A); however, the functional consequence of the phosphorylation events is the same. Surprisingly, fission yeast Ckb1 has a negative effect on the catalytic activity of Cka1 on fission yeast PC4 and its presence inhibits the PC4 phosphorylation, therefore preventing the inhibition of PC4 function by Cka1. It is well known that the regulatory subunit (Ckb1) of CK2 has stimulatory activity on the CK2 substrates, and only a few of them have been reported to be downregulated by the regulatory subunit. Substrates that are negatively regulated by the CK2β subunit are unique, such as calmodulin [53]. This work provides an example of the negative regulation of transcriptional cofactor activity by CK2 phosphorylation in fission yeast, which is antagonized by the regulatory Ckb1 subunit. Furthermore, fission yeast PC4 has been found heavily phosphorylated in vivo at different serine and threonine residues, especially Ser96 and Ser98 during the cellular response to thiabendazole [54] and Ser98 and Ser100 during the mitotic M phase [55]. In addition, Ser96 and Ser98 are in vivo phosphorylated during the response to nitrogen starvation [56].

In fission yeast Cka1 is essential for cell viability [52], however, deletion of Ckb1 causes a cold-sensitive phenotype and abnormal cell shape, while its overexpression causes cell growth inhibition and cytokinesis [57]. The involvement of CK2 in transcription has been well documented, and it can regulate transcription by all three eukaryotic RNA polymerases [28,58]. In mammals, CK2 is able to phosphorylate several GTFs of the RNA polymerase II transcription system, leading to the increment of transcription activity [30], and also vertebrate CK2 is able to modulate the activity of fission yeast TBP, reducing the binding of this transcription factor to the TATA box [59]. Furthermore, in fission yeast it has been demonstrated that CK2 negatively regulates the binding of the transcription factor Rrn7 to the HomolD box of ribosomal protein genes both in vitro and in vivo [60]. Therefore, CK2 is an important regulator of RNA polymerase II transcription in higher eukaryotes and in fission yeast as well.

In conclusion, our results indicate that fission yeast PC4 can be negatively regulated by phosphorylation at Ser98 and Ser100 by the catalytic subunit (Cka1), while the regulatory Ckb1 subunit has a negative effect on the phosphorylation of fission yeast PC4 by Cka1. Those phosphorylation sites were identified in vitro, but they might be important in vivo since those residues are found phosphorylated in vivo as well. This indicates that some extracellular and/or intracellular signals could signal CK2 to regulate RNA polymerase II transcription through PC4 or other transcription factors.

## 4. Materials and Methods

### 4.1. Expression and Protein Purification of CK2α and CK2β from S. pombe (Cka1 and Ckb1)

We designed specific primers with restriction site adaptors to amplify the complete ORF of CK2α and CK2β (Cka1 and Ckb1, respectively) using the sequences available in Genbank. Both ORFs were amplified from a fission yeast cDNA library. The accession numbers of these sequences are NM_001019073 for CK2α and NM_001020034 for CK2β and were cloned as described previously [57]. These sequences were cloned into pGEMT-Easy (Promega, Madison, WI, USA) and then were subcloned into the NdeI/BamHI restriction sites of the expression vector pET15b (Novagen, Madison, WI, USA). *E. coli* BL21 (DE3) cells were transformed with pET15b containing the corresponding gene sequences, grown in LB media until OD_600_ 0.8 and then the protein expression was induced with 0.5 mM IPTG (Calbiochem, USA) to obtain appropriate amounts of those recombinant proteins. To recover the recombinant proteins, first the cell pellets were washed with STE buffer (100 mM NaCl, 10 mM Tris HCl pH 8.0, 1 mM EDTA). Then, mixes were sonicated and centrifuged to collect the pellets, which were washed with lysis buffer (50 mM HEPES pH 7.9, 5% glycerol, 2 mM EDTA, 0.1 mM DTT, 0.05% DOC, 1% Triton X-100) and then with the same buffer but without detergents. The washed pellets were mixed with 40 mL guanidine buffer (6M guanidine hydrochloride, 10 mM HEPES pH 7.9, 0.2 mM EDTA, 0.2 mM EGTA, 2 mM DTT) and were incubated overnight at 4 °C. The next day, the protein mixes were diluted with 160 mL dilution buffer (10 mM HEPES pH 7.9, 0.2 mM EDTA, 0.2 mM EGTA, 2 mM DTT) and were incubated overnight at 4 °C. Then, the mix was dialyzed for 4 h at 4 °C against 20 mM Tris pH 7.5, 0.5 M KCl, 20 mM imidazole, 10% glycerol (30% glycerol for Cka1), 0.01% Triton X-100, 1 mM β-mercaptoethanol and 0.1 mM PMSF. The protein mix was centrifuged to discard denatured proteins and the supernatant was saved for Ni-NTA-Agarose resin purification under native conditions. Then, 100 mL of protein mix was passed through 1 mL of settled Ni-NTA-Agarose resin ProBond (Invitrogen, Waltham, MA, USA), previously equilibrated with the same dialysis buffer. The resin was then washed until no proteins were detected in the flow-through. Afterwards, to recover the recombinant proteins, 200 mM imidazole was used to elute the protein from the resin. The eluted proteins were analyzed in SDS-PAGE followed by silver staining.

### 4.2. Recombinant Fission Yeast PC4 Expression and Purification

Expression of recombinant fission yeast PC4 in *E. coli* BL21 (DE3) was performed according to Contreras-Levicoy et al., 2008 [11]. Briefly, PC4-expressing cells were grown in 500 mL of terrific medium at 37 °C until OD_600_ of 0.8. Cultures were induced with 0.5 mM IPTG for 4 h. Cells were harvested by centrifugation at 3000× *g* for 10 min. Pellet was resuspended in 20 mL of 7M urea and sonicated. Then, recombinant proteins were purified using Ni-NTA-Agarose resin following the manufacturer’s instructions (Invitrogen, USA). After purification, PC4 protein was renatured by dialysis against 20 mM HEPES pH = 7.5, 100 mM KCl, 0.1 mM EDTA, 5 mM DTT, 10% glycerol and 0.1 mM phenylmethanesulfonyl fluoride. Mutant genes of fission yeast PC4 were synthesized at Genscript, Inc. (Piscataway, NJ, USA), by changing the Ser98 and Ser100 codons to Ala encoding codons and inserting them into pET15b. Mutant proteins were expressed and purified as described previously.

### 4.3. CK2 Kinase Activity Assays

Kinase assays were performed in a final volume of 25 μL, containing 50 mM HEPES 7.9, 75 mM KCl, 7 mM MgCl_2_, 0.5 mM DTT, 50 μM ATP, 0.5 μCi [γ^32^P] ATP, different amounts of Cka1 (0.13, 0.25, 0.5, 1, 2 and 4 pmol) and Ckb1 (0.25, 0.5, 1, 2 and 4 pmol) and casein (1 μg) or PC4 (1 and 10 μg) as substrates. In inhibition assays, reactions were performed with Cka1 (2 pmol) and PC4 (1 μg) as substrate and increasing amounts of heparin (0.25, 0.5, 1, 2 and 4 μg/mL). All the assays were incubated for 30 min at 30 °C. The reactions were stopped by adding an SDS-PAGE sample buffer and the assays were analyzed by SDS-PAGE (15% polyacrylamide gel). The gels were dried and exposed to X-ray film (Kodak, Rochester, NY, USA) at −80 °C for 30 min and then revealed in a dark room.

### 4.4. In Vitro Phosphorylation Assays

Mutants and wild type PC4 (5 μg of protein) were incubated in a reaction buffer containing 0.5 mM ATP and 100 pmol of Cka1 for 30 min at 30 C. Afterwards, the reaction was loaded onto an S-300 gel filtration column (30 × 1 cm) equilibrated in 50 mM HEPES pH 7.9, 1 mM EDTA, 2 mM DTT, 100 mM KCl and 0.2 mM PMSF, in order to separate the kinase and PC4. The fractions containing PC4 were quantified and used in transcription assays. Some reactions were performed in the presence of 100 pmol of Ckb1 and processed as indicated.

### 4.5. In Vitro Transcription Assays

In vitro transcription reactions were performed according to Tamayo et al., 2004 [61]. For transcription assays we used fission yeast whole cell extracts and the Ad-MLP fused to the G-less cassette as a template (pG5MLP). In each reaction, 50 ng of template was used in order to obtain basal transcription.

### 4.6. EMSA

EMSA experiments were performed according to Tamayo et al., 2002 [62]. Briefly, proteins were incubated with 0.1 ng of labeled DNA probe containing the sequence −20 to +40 of the Ad-MLP. Proteins were incubated for 45 min at 30 °C. Complexes were analyzed in a 6% PAGE gel containing Tris-borate EDTA buffer (TBE) at pH = 8.2 (40 mM Tris, 40 mM boric acid, 1 mM EDTA) supplemented with 4% glycerol. The gels were dried and exposed to X-ray film (Kodak, USA) at −80 °C for 24 h and then revealed in a dark room.

### 4.7. Statistics

GraphPad Prism 9 (GraphPad Software Inc., San Diego, CA, USA) was used for all the analyses. Results were expressed as the mean ± standard deviation (SD). All experiments were repeated three times. Analyses of autoradiography films were performed using the Image J software 1.53k version (NIH, Bethesda, MD, USA). Distribution of data was evaluated by Shapiro–Wilk test and differences between experimental groups were evaluated using the Student’s *t*-test; *p* < 0.05 was considered as statistically significant.

## Figures and Tables

**Figure 1 ijms-23-09499-f001:**
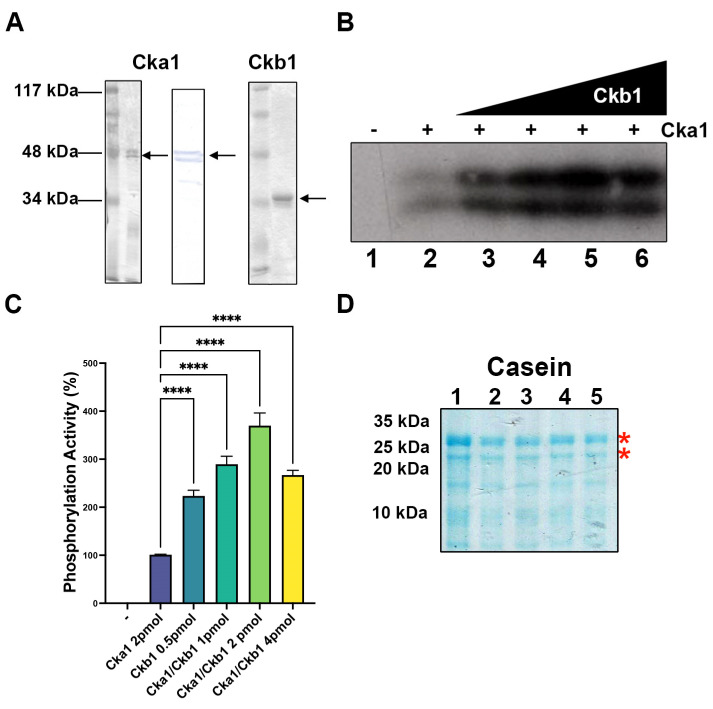
Purification and activity of fission yeast Cka1 and Ckb1. (**A**) Silver-stained PAGE-SDS and a Western blot (Cka1) of the purified subunits Cka1 and Ckb1. A 12% gel was used to separate the proteins and 200 ng of each subunit was analyzed. (**B**) Casein phosphorylation was determined using 2 pmol of Cka1 and increasing amounts of Ckb1 from 0.25 to 4 pmol (lanes 2–6)—indicates assay without Cka1 (lane 1). (**C**) The signal from each lane from (**B**) was quantified and plotted using the Image J software as described in Materials and Methods; – indicates a negative control without Cka1 and + indicates the assays with the addition of the enzyme Cka1. (**D**) A 14% PAGE-SDS gel was used to analyze the casein substrate used in (**B**) and shows that all lanes contain the same amount of substrate. Red asterisks show the phosphorylated polypeptides indicated in (**B**); **** indicates *p* < 0.0001.

**Figure 2 ijms-23-09499-f002:**
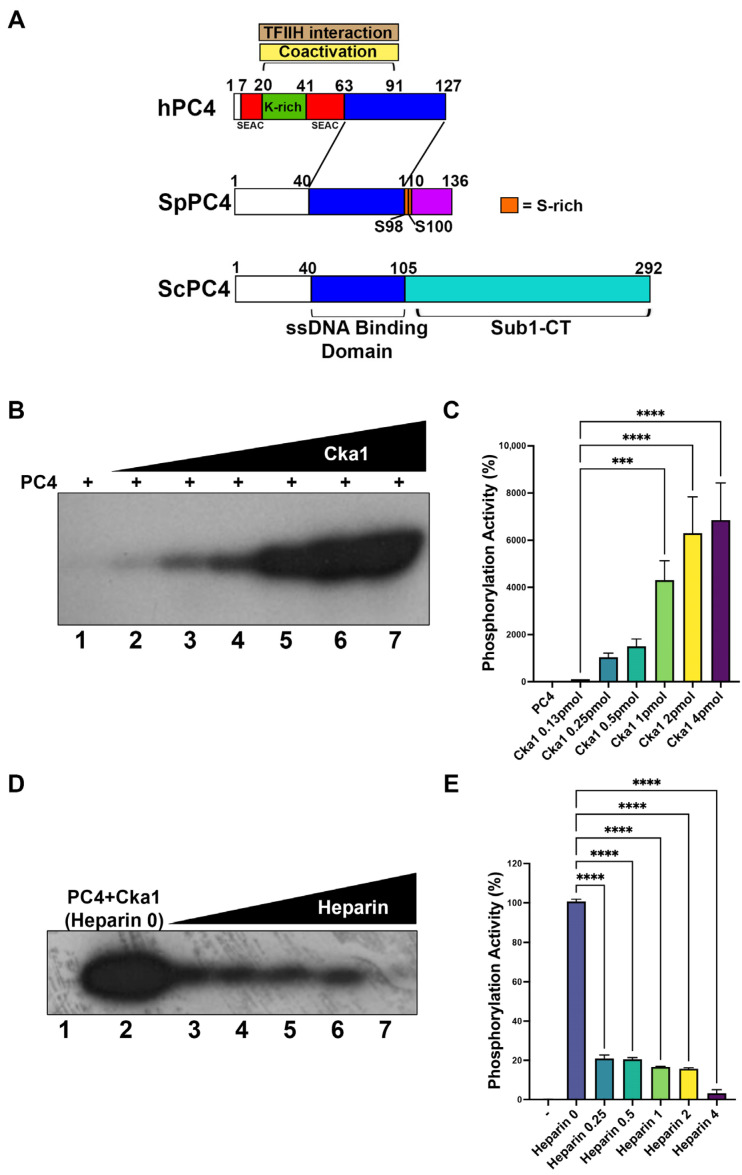
Fission yeast Cka1 phosphorylates fission yeast PC4. (**A**) Schematic representation of human PC4 (hPC4), fission yeast PC4 (SpPC4) and yeast PC4 (ScPC4; Sub1) that shows the extension of the known structural domains. Human PC4 possesses two SEAC domains at the N-terminal portion, which are targets for CK2 phosphorylation. It also possesses a K-rich domain at the N-terminus and a TFIIH domain which overlaps the coactivation domain. All three possess a single-stranded DNA binding domain and at the same location the dimerization domain. Yeast Sub1 possesses an extra C-terminal domain of unknown function (purple region). The location of the mutated Ser residues of SpPC4 are shown (S98 and S100). (**B**) SpPC4 phosphorylation by Cka1. PC4 (500 ng) was phosphorylated with increasing amounts of Cka1 (lanes 2–7; 0.1, 0.25, 0.5, 1.0, 2.0 and 4.0 pmol of fission yeast Cka1). Negative lane 1 only contains PC4; + symbols indicate the addition of PC4 to the assays. (**C**) Signals of each lane from (**B**) were quantified and plotted using the Image J program. (**D**) Heparin at 0.25, 0.5, 1, 2 and 4 μg/mL (lanes 3–7) was used to inhibit the activity of Cka1. (**E**) Lanes from (**D**) were quantified and plotted using the Image J software; *** indicates *p* < 0.001, **** indicates *p* < 0.0001.

**Figure 3 ijms-23-09499-f003:**
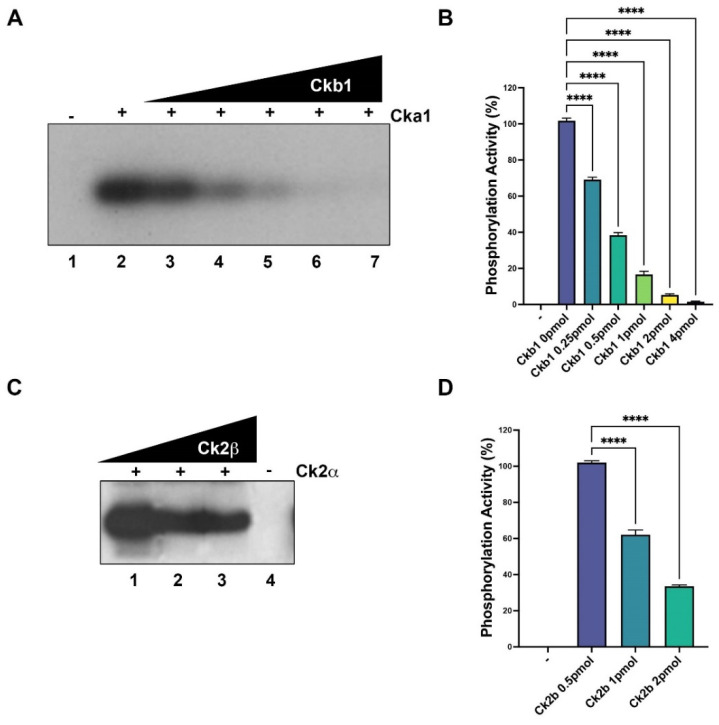
Fission yeast Ckb1 modulates the catalytic activity of Cka1. (**A**) Cka1 (2 pmol) was used to phosphorylate fission yeast PC4 (500 ng) in the presence of increasing amounts of fission yeast Ckb1 (0.25, 0.5, 1.0 and 2.0 pmol; lanes 2–7, respectively); – indicates an assay without Cka1 and + indicates the assays with the addition of Cka1. (**B**) Signals of each lane from (**A**) were quantified and plotted using the Image J software. (**C**) CK2α (2 pmol) from *X. laevis* was used to phosphorylate fission yeast PC4 in the presence of 0.5, 1.0 and 2.0 pmol of X. laevis CK2β (lanes 1–3). Lane 4 contains fission yeast PC4 and CK2β; – indicates a control without enzymes and + indicates the assays with the addition of Cka1. (**D**) Signals of each lane from (**C**) were quantified and plotted using the Image J software; – indicates assays without Cka1, but in the presence of PC4; **** indicates *p* < 0.0001.

**Figure 4 ijms-23-09499-f004:**
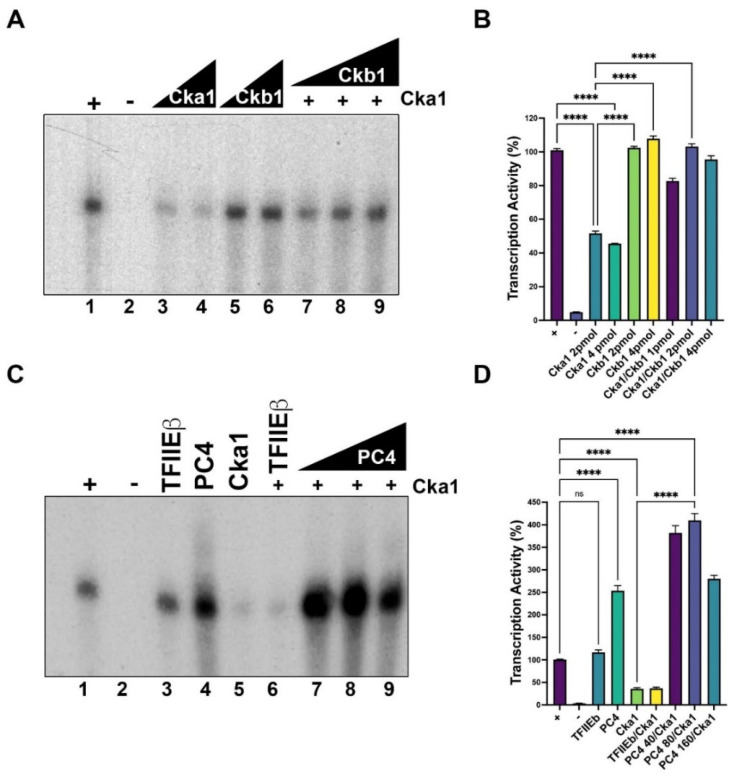
Fission yeast Cka1 and Ckb1 can modulate basal transcription in a fission yeast whole cell extract. (**A**) Basal transcription (lane 1), labeled (+) in the presence of Cka1 (2.0 and 4.0 pmol, lanes 3 and 4, respectively), can downregulate basal transcription from the Ad-MLP promoter. Ckb1 (2.0 and 4.0 pmol, lanes 5 and 6) added to the transcription assay has no effect on transcription. CKa1 (4 pmol) in the presence of Ckb1 (1.0, 2.0 and 4.0 pmol, lanes 7–9l) restores basal transcription. The lane 2 labeled (−) represents the negative control without DNA template. (**B**) The signals of each lane from (**A**) were quantified and plotted using the Image J software. (**C**) Lane 1 (+) represents basal transcription and the lane labeled TFIIEβ represents the basal transcription with this polypeptide added, while the lane labeled PC4 represents basal transcription in which PC4 was added. Cka1 was added to the basal assay in the lane 5 labeled Cka1. Lane 6 labeled +TFIIEβ represents an assay in which Cka1 and TFIIEβ were added simultaneously. Fission yeast PC4 (40, 80 and 160 ng, lanes 7–9) was added in those lanes labeled PC4, in the presence of 4 pmol of Cka1. Lane 2 labeled (−) represents a negative control without template. (**D**) Those signals of each lane from (**C**) were quantified and plotted using the image J software; ns indicates no significative, **** indicates *p* < 0.0001.

**Figure 5 ijms-23-09499-f005:**
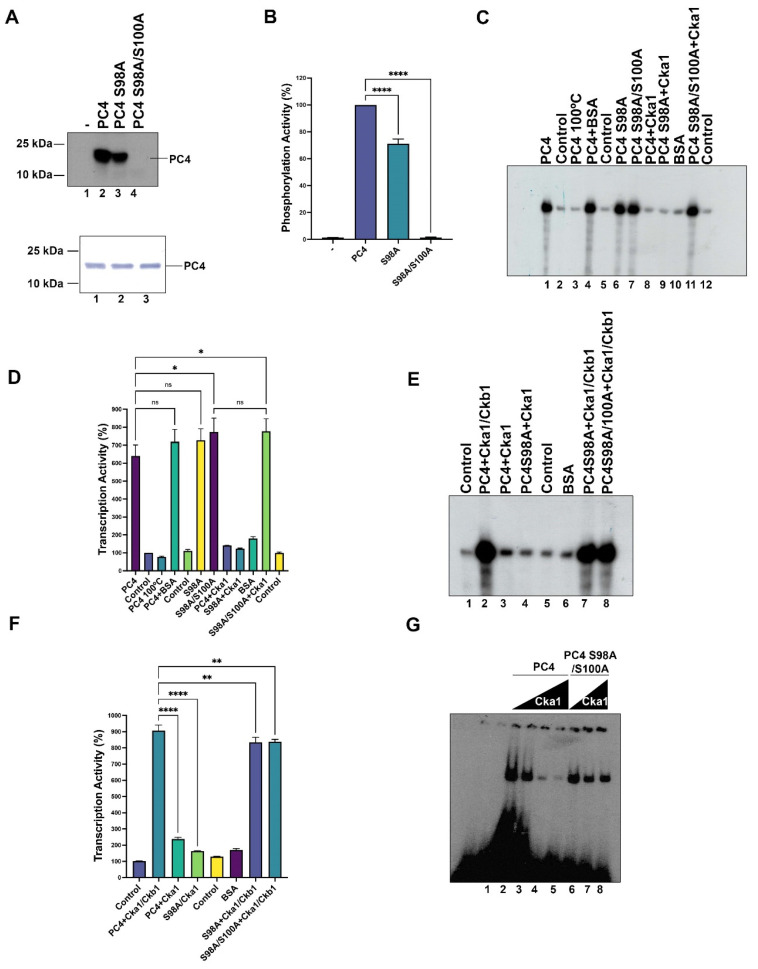
Serine residues S98 and S100 are important for fission yeast PC4 function. (**A**) Phosphorylation of wild type, S98A and S98A/S100A PC4 polypeptides (500 ng) with 10 pmol of fission yeast Cka1, as described in Materials and Methods. Lower panel shows a Western blot of the different proteins (100 ng) used in the upper panel (lanes 1–3). (**B**) The signals of each lane from (**A**) were quantified and plotted using the Image J software. (**C**) Wild type PC4 and mutants (40 ng) were assayed in transcription assays using fission yeast whole cell extracts adding wild type PC4 or combinations of Ck1a (4.0 pmol) or BSA (4.0 pmol), as indicated on the top of the figure. The lanes 2, 5 and 12, labeled control, indicate basal transcription from the whole cell extract in the absence of exogenous proteins. PC4 100 °C indicates that PC4 was inactivated by heat. (**D**) The signals of each lane from (**A**) were quantified and plotted using the Image J program. (**E**) The effect of CKb1 on Cka1 activity on the transcriptional functions of wild type and mutant PC4 polypeptides was measured in the transcription assay using different proteins and combination of proteins, as indicated at the top of the panel. Control (lanes 1 and 5) indicates basal transcription in which no exogenous proteins were added to the assay. In this assay the amounts of PC4 polypeptides and Cka1 were the same as used in (**C**). (**F**) The signals of each lane were quantified and plotted using the Image J program, as described in Materials and Methods; ns indicates no significative, * indicates *p* < 0.05, ** indicates *p* < 0.01, **** indicates *p* < 0.0001. (**G**) EMSA assay showing that phosphorylation of PC4 by Ck1a inhibits its dsDNA binding activity. In this assay, 40 ng of wild type PC4 was incubated with dsDNA oligonucleotide and increasing amounts of Cka1 (0.5, 1 and 2 pmol; lanes 3–5, respectively) were added in the presence of 0.5 mM ATP. In addition, the PC4 mutant S98A/S100A (40 ng) was assayed in the presence of ATP and increasing amounts of Cka1 (1 and 2 pmol, lanes 7 and 8, respectively) were added. Lanes 2 and 6 represent controls in which Cka1 was not added. Lane 1 does not contain protein.

## Data Availability

Not applicable.

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
