# Peer review of "The Catalytic Subunit of Schizosaccharomyces pombe CK2 (Cka1) Negatively Regulates RNA Polymerase II Transcription through Phosphorylation of Positive Cofactor 4 (PC4)"

_ijms, 2022, doi:10.3390/ijms23169499_

Round 1
Reviewer 1 Report
The manuscript by Rojas et al., demonstrates that fission yeast PC4, which is an evolutionarily conserved transcription coactivator, is phosphorylated by CK2 at least under in vitro conditions. They were able to identify the sites of CK2-mediated phosphorylation of PC4 as serine-98 and serine-100. CK2-mediated phosphorylation of PC4 negatively regulated its transcription activity in an in vitro transcription assay using Ad-MLP. These results are reminiscent of mammalian PC4, which is also phosphorylated by CK2 in cells and phosphorylation inhibits coactivator function of the protein. The results describe in this manuscript give rise to the speculation that fission yeast PC4 may be similarly regulated by CK2-mediated phosphorylation under in vivo conditions. The following suggestions will, however, led further credence to author’s hypothesis.
(1) The study will get a boost if authors can demonstrate that fission yeast PC4 is a phosphoprotein even under in vivo conditions. Authors have immense experience in protein purification. They can purify fission yeast PC4 and find sites of phosphorylation by mass spectrometry. They may find more phosphorylation sites in addition to serine-98 and serine-100 as kinases other than CK2 may also be phosphorylating PC4 in fission yeast cells.
(2) Authors performed experiments with mutant PC4 with serine-98 and serine-100 mutated to alanine residue. Authors can similarly make phosphomimetic mutants by mutating serine-98 and serine-100 to aspartic. If author’s hypothesis is correct, phosphomimetic mutants will exhibit reduced activity in the in vitro transcription assay with Ad-MLP.
(3) In all figures with gel data, lane numbers must be indicated under each lane. Author’s refer to lane numbers in the text, but there are no lane numbers marked in figures.
(4) Manuscript has a number of grammatical errors. These must be rectified before re-submitting the manuscript.
Author Response
We thanks to the reviewer 1 for the useful comments on the manuscript. Certainly, those comments will improve the quality of the manuscript. Please, you can find below our reply to your comments.
- To determine in vivo phosphorylation, we have planned to use a PC4-TAP tagged strain to purify native PC4 and PC4-associated proteins and identify them by mass spectrometry. We have not attempted to purify it directly from whole cell extracts, since we learnt from our work that fission yeast whole cell extracts have potent phosphatases, which might dephosphorylate most of the PC4, even though using a powerful cocktail of phosphatase inhibitors. We know that because in our attempts to purify phosphorylated RNA pol II (IIo) by using conventional chromatography, we have failed due to the high phosphatase activity present in the cell extracts. However, literature reports have identified PC4 as a phosphoprotein, precisely in S96, S98 and S100. Those references have been included in the discussion section (references 54 – 56).
- We mutated PC4 in S98 and S100 to Ala residue, since this amino acid residue might not change the 3D structure of the molecule. If we change those serine residues to aspartic to make phosphomimetic mutant, we expect that those mutants might not be able to stimulate transcription any longer. Unfortunately, making mutants, expressing, purifying, and assaying the mutants in a transcription assay takes a long time and we have only ten days to resubmit the manuscript according to the IJMS instructions.
- We have added lane numbers in the figures according to the reviewer comments.
- We have revised all grammatical mistakes and improved the language according to the reviewer comment. We apologize for those mistakes.
Reviewer 2 Report
Mammalian PC4 is a ssDNA binding protein and a well-known transcription coactivator, which is conserved in budding and fission yeast (Sub1), as well as in other eukaryotic and even prokaryotic organisms. PC4 and Sub1 have been shown to activate transcription in vitro, and in the case of human (PC4) and S. cerevisiae (Sub1), several studies have shown their roles in different aspects of gene expression, not only in vitro but also in vivo. In S. pombe, Sub1 can also stimulate basal and activated transcription, as human PC4 does.
PC4 phosphorylation has been studied in mammals and showed that it inhibits the coactivation function, then negatively regulating transcription. In the case of budding yeast, there is a report showing that Sub1 phosphorylation in vitro may alter its transcriptional functions. Thus, Henry et al, (1996) showed that phosphorylated and unphosphorylated Sub1 have different properties, such as unphosphorylated Sub1 interacts with the general transcription factor TFIIB whereas phosphorylated Sub1 does not, or that unphosphorylated Sub1 interacts with Gal4-VP16 more strongly than does phosphorylated Sub1. These binding properties therefore changes Sub1 transcription activation functions.
Rojas et al, describe in their work the effect of fission yeast PC4 phosphorylation by CK2 in basal transcription in vitro. CK2 consists of regulatory and catalytic subunits, Ckb1 and Cka1, respectively. The authors show that CK2 phosphorylates PC4 in vitro. Cka1 displays an inhibitory effect in basal transcription by phosphorylating PC4 whereas Ckb1 is able to relieve the inhibitory effect. The authors also identified PC4 phospho-sites, that apparently are the targets of Cka1 kinase activity. Although the data are convincing, in my opinion, a major revision should be done in order the manuscript to be accepted for publication.
Major comments
The authors do not provide any in vivo data corroborating or supporting the functional relevance of their in vitro results. Therefore, the work presented is merely descriptive, and do not add further significative advance to the field, other than Sub1 in S. pombe, as in mammals, can be phosphorylated by the same kinase. The authors have the tools to have tried to find out what is the in vivo role of Sub1 phosphorylation in RNAPII transcription, according to the title of the manuscript and their conclusions. For instance, to investigate whether the identified residues could affect RNAPII recruitment to genes during transcription activation, or if altering CK2 activity or regulation could affect Sub1 association to gene promoters during transcription initiation.
Does Sub1 phosphorylation by CK2 affects constitutive or induced transcription?.
Does Sub1 phosphorylation by CK2 affect transcriptional activation in S. pombe?
Does Sub1 phosphorylation affects ssDNA binding affinity, as in the case of PC4 and budding yeast Sub1, and therefore transcription?. An EMSA assay could be an easy and good experiment to answer to this. .
To answer to any of these questions will improve the manuscript, will add further significative advance to the field and will support the authors’ conclusions.
The in vitro transcription assays have been done using cell extracts where I guess in addition to endogenous PC4 and CK2, other kinases are present that could affect PC4 phosphorylation and therefore in vitro transcription results. This should be commented when discussing the results.
During the purification of Cka1 two bands are obtained (Figure 1A), with no further comment whether this could be due to a contamination with some other protein/kinase. Do they authors corroborate the identity of the kinase, though they indirectly showed that it could be when adding heparin to the kinase?. Nevertheless, to investigate if the two bands correspond to Cka1, or instead they have a contamination that could influence the results, will improve the presented data, or at least should be commented.
The authors show a positive control of Cka1 functionality using casein as a known target of this kinase, but I miss some negative control to corroborate the specificity of Cka1 kinase activity over PC4. It is known that purified kinases tend to be promiscuous in in vitro reactions.
The authors used Ɣ32P-ATP in the kinase assays, but then they quantified the gels signal with Image J software, instead of using a phosphorimager or similar equipment. How are they sure that the bands are not saturated? For instance, in Figure 2B or 2C, the bands in the corresponding lanes are not separated enough to do the quantification.
The Discussion should be improved. The first part of the Discussion will fit better in the Introduction section (mostly the two first paragraph). There are few discussions regarding the role of in vivo Sub1 phosphorylation in fission yeast. They do not discuss about the fact that hPC4 is phosphorylated by CK2 in the N-terminal region and that S. pombe PC4 in the C-terminus, and if other residues localized in similar position as in PC4 are could be phoshosites. I suppose that other phosphosites were found in the search the authors did.
Figure 4.
How sure we can be that there is not an excess of PC4 that Cka1 cannot phosphorylate and therefore it is activating transcription? As I understand phosphorylated PC4 by Cka1 is unable to activate transcription (Figure 5C and E).
The authors do not pay attention to the fact that transcription is repressed when PC4 levels are increased (Werner et al, 1998). How this fits with the results in the assay shown in figure 4C?. In fact, lane 9 in this figure support it.
The loading controls for the different assays are missing. For instance, in figures 2B, 2D, 3A, 5A. A western blot with anti-HIS to show that in all lanes there is the same amount of PC4 is quite feasible because the recombinant protein is HIS-PC4.
Minor comments
In general, the quality of the images is not good enough. The images are quite small. For instance, the plot legends are so tiny that it is hard to read them. Figure legends some times are not enough explicative.
It will be very helpful for the readers it the authors add the number of the lanes below the gel images (figures 4A, 4C, 5C and 5E), because they refer all the time to lines number. In addition, to label the gels indicating to what correspond the bands will improve the images, for instance figures 3A and C.
Similarly, in figure 1B, what are the two bands?. Both bands correspond to phosphorylated casein? It will be good to indicate it in the figure itself. A western blot or stained gel is missing to show that the amount of casein is similar in all the lanes.
Lane 81 says: ” It is unknown whether PC4 is phosphorylated or not by CK2 in budding yeast and what are the effects of such modification”. As commented above, Henry et al, 1996, showed that phosphorylated and unphosphorylated PC4 have different properties. I think this should have been mentioned here or even in the discussion.
Lane 84: S. cerevisiae PC4 has already a name, Sub1. It will be much better to use this name along the text. In the case of S. pombe, they should decide how to name it, but being consistent (Sub1, sub1 or fission yeast PC4….).
Figure 2. “Fission yeast Cka1 phosphorylates fission yeast PC4, and it is modulated by Ckb1”. This figure has nothing to do with Ckb1.
Lane 165-166. “We found that Cka1 phosphorylates PC4 and this activity was increased according to the amount of Ck1a added (figure 2B and C)”. Better say that the level of PC4 phoshorylation increased according to the amount of Ck1a. The gel shows PC4 phosphorylation.
Lane 188: “compare Figure 3C, lane 6 with Figure 3B”. There is no lane 6 in figure 3C.
There are two Figure 3C. Should be the right panel corrected by 3D?
Figure 5A.
I see a contradiction between figure 5E and figure 4, comparing lane 3 and lanes 7,8 and 9, respectively. Adding PC4+Cka1 inactivates transcription in one case, and in the other activates it. I think the experiments are not well explained.
Text from 249 to 251 is confusing. It seems that both, 100ºC and BSA treatments have the same effect on transcription activation, and in the first case it denaturalizes PC4, and in the second one, the BSA does not increased the transcription further than PC4 itself.
I think the text sometimes is confusing and this make difficult to understand the result.
Missing References:
288: There is a recent published work proposing a role for the CT of budding yeast Sub1.
300-301: the reference for Sub1 and Rna15 interaction in the modulation of transcription termination is missing.
304: Reference 46 is about PC4 and mammalian cells, is not a yeast Sub1 paper.
To use as reference the Pombase is not a good choice. I think it will be much better to add references from published and peer reviewed studies.
Author Response
We thanks to reviewer 2 for all the comments on the manuscript. Those series of comments will certainly improve the quality of the manuscript. All answers to your queries can be found below.
- We have mainly reported in vitro results, but the evidence points that those residues are important in vivo, since PC4 is a phosphoprotein, which can be in vivo phosphorylated at several residues and precisely S96, S98 and S100 are the major phosphorylation sites. The references have been included in the Discussion section (references 54 – 56). The importance of our work is the identification of those sites as targets for protein kinase CK2 and the demonstration that those sites are important for PC4 function. Fission yeast PC4 is not an essential gene for viability and its deletion do not cause any detectable phenotype, thus, genes whose transcription is affected for PC4 have been not described yet. We believe that PC4 might be important for the transcription of a set of genes, which are not essential for yeast growth.
2. Whether or not fission yeast PC4 can affect constitutive or activated transcription is still unknown. We can speculate that can affect both, but we do not have in vivo data to make a definitive conclusion yet. Our in vitro data indicates that can affect basal transcription.
3. Unfortunately, we did not perform EMSA assays to assay the ssDNA binding activity of wild type phosphorylated fission yeast PC4 and the mutants. However, in an earlier work, we demonstrated that recombinant PC4 can bind ssDNA. Fortunately, we have done EMSA assays using dsDNA to analyze the binding activity of fission yeast PC4 and we found that phosphorylated PC4 weakly binds to dsDNA, compared to the unphosphorylated PC4, while the doble mutant S98/S100 is not affected for Cka1 phosphorylation. That experiment was included in the Results section (Figure 5G). Those results indicate that the dsDNA binding activity is inhibited by phosphorylation and therefore its transcriptional activity should also be inhibited.
4. Purified Cka1 contains two polypeptides, and both are Cka1 as confirmed by western blot with antibodies against Xenopus CK2α. The western blot was included as Figure 1A. Is important to note that those are recombinant proteins expressed in E. coli and Ser/Thr kinases have not reported in bacteria. Cka1 specificity on PC4 can be seen in Figure 2B and additionally in Figure 2D when reactions containing PC4 without Cka1 does not show any reaction. The experiments were done in triplicate to quantify and plotted them by using the image J program. We were not able to use a Phosphorimager or a similar equipment, because the Facultad de Medicina at the Universidad de Chile does not possess such equipment. We apologize for that, but every time we use 32P isotopes, we must use films and a program to scan and quantify the signals. In addition, to avoid saturation of the marks in the films, we checked different exposure times before analysis.
5. We have improved Figure 4, and we further discuss those results according to the reviewer suggestion.
6. We measured protein concentration by the Bradford assay, using BSA as standard, and we used the very same lot of PC4 proteins in all the reported experiments. Wild type and mutant PC4 were more than 95 % pure as judged by PAGE-SDS followed by Coomassie blue R-250 staining (data not shown in the manuscript). All of them were adjusted at the same concentration as it can be seen from a western blot in Figure 5A (bottom panel). We believe it would not be necessary to show the same control in each experiment involving PC4. We have enlarged the figures and put numbers in each lane of the figures as suggested for the reviewers. Also, text and figure legends were given more explanation and we tried to clarify every issue that was not enough explicative. Also, a loading control for casein was added (Figure 1D)
7. Lane 81, we rectified it, we meant “the protein kinases that can phosphorylate Sub1 are unknown yet”. We have included that in the text. Lane 84, we have used the name fission yeast PC4 along the entire text. Figure 2 was fixed, and we apologize for that error. Lane 165-166, we have changed the sentence according to the reviewer suggestion. Lane 188, we have corrected that, and we apologize for the mistake.
8. Figure 5A. We clarified and relabeled the figure and corrected text and figure legends. We hope now can be clear enough. Text from 249-251 has been corrected to make it clear for the readers.
9. All missing references were added, and we again apologize for the mistakes. We changed the references of Pombase for references from published studies, although we think that we should not be afraid to use as reference the Pombase, since it is a big effort from the Pombase team to keep all the information that can be mined easily, and they deserve some kind of recognition. Lastly, we discuss the possible role of the CT domain of budding yeast Sub1, according to the work from Calvo and colleagues. We were not able to find any recent work on this issue.
Round 2
Reviewer 1 Report
I agree with response of authors.
Author Response
Dear Reviewer 1:
Thank you very much for your comments that helped us to improved the manuscript.